# Phenotypic Investigation and RNA-seq of *KN1* Involved in Leaf Angle Formation in Maize (*Zea mays* L.)

**DOI:** 10.3390/ijms25063180

**Published:** 2024-03-10

**Authors:** Yuanming Wu, Yunfang Zhang, Zelong Zhuang, Xiangzhuo Ji, Jianwen Bian, Jinhong Xian, Yinxia Wang, Yunling Peng

**Affiliations:** 1College of Agronomy, Gansu Agricultural University, Lanzhou 730070, China; 2Gansu Provincial Key Laboratory of Aridland Crop Science, Gansu Agricultural University, Lanzhou 730070, China; 3Gansu Key Laboratory of Crop Improvement & Germplasm Enhancement, Gansu Agricultural University, Lanzhou 730070, China

**Keywords:** maize (*Zea mays* L.), mutant *kn1*, leaf angle, RNA-seq, exogenous phytohormones, weighted gene co-expression network analysis

## Abstract

Leaf angle (LA) is one of the core agronomic traits of maize, which controls maize yield by affecting planting density. Previous studies have shown that the *KN1* gene is closely related to the formation of maize LA, but its specific mechanism has not been fully studied. In this study, phenotype investigation and transcriptomic sequencing were combined to explore the mechanism of LA changes in wild type maize B73 and mutant *kn1* under exogenous auxin (IAA) and abscisic acid (ABA) treatment. The results showed that the effect of exogenous phytohormones had a greater impact on the LA of *kn1* compared to B73. Transcriptome sequencing showed that genes involved in IAA, gibberellins (GAs) and brassinosteroids (BRs) showed different differential expression patterns in *kn1* and B73. This study provides new insights into the mechanism of *KN1* involved in the formation of maize LA, and provides a theoretical basis for breeding maize varieties with suitable LA.

## 1. Introduction

Maize (*Zea mays* L.) is one of the most widely grown food crops around the world, providing the largest amount of food production and being used on a large scale as staple food, animal feed [1], and chemical raw materials [2,3,4,5]. Therefore, cultivating maize varieties with excellent agronomic characteristics is a necessary measure to deal with the increasingly acute problems of population, farmland and food [6,7]. Maize yields per unit of land in the United States, the world’s largest maize producer, have increased more than sevenfold since the 1930s [8], largely because of increased planting density.

Plant architecture is an important factor restricting crop yield. The ideal plant architecture is beneficial to increase planting density and produce more grain with limited cultivated land resources. In maize, plant architecture is determined by many factors such as leaf angle (LA) and plant height (PH) [9], which are essentially quantitative traits controlled by environmental factors and multiple genes [10,11]. Appropriate LA and PH can increase the planting density and yield by increasing the light capture area of leaves and reducing lodging, respectively. As endogenous growth signals, phytohormones play an important role in regulating plant growth and development [12]. A number of studies have confirmed that auxin (IAA), brassinosteroids (BRs), gibberellins (GAs) and strigolactones (SLs) are involved in regulating LA. Studies on a series of IAA synthesis-deficient rice mutants with increased LA showed that IAA promoted a reduction in LA [13,14,15], and related studies on P-glycoprotein ZmPGP1 confirmed that IAA was also involved in the regulation of LA in maize [16,17]. Similar to IAA, rice mutants with deficient SL synthesis also have increased LA [18], and SLs may be involved in regulating LA with other phytohormones [19]. GAs have a dual function: endogenous GAs usually promote the increase of LA [20], while exogenous GAs have the opposite effect [21]. Compared with the above phytohormones, BRs are at the core of LA regulation. They play a role in promoting the increase in LA in maize and rice [22,23], and can also interact with a variety of phytohormones and participate in the regulation of LA [19,20,24]. In addition, a recent study on maize mutant *Semidwarf3* (*Sdw3*) confirmed that ethylene also plays an important role in the regulation of LA [25]. In contrast to the abovementioned phytohormones whose functions and mechanisms have been clearly elucidated, there are few reports that abscisic acid (ABA) is directly involved in LA formation. A study using three different rice cultivars and their corresponding mutants showed that ABA has antagonistic effects on the biosynthesis and signal transduction of BRs, thus leading to a reduction in LA by disrupting the biological function of BRs [26]. Further studies are needed to prove whether ABA has other functions and mechanisms in regulating LA.

KNOTTED1-like homeobox (*KNOX*) genes encode plant-specific homeobox transcription factors (TFs). This gene family belongs to the three amino acid length extension (TALE) superclasses of homologous proteins [27]. Typical KNOX proteins contain four highly conserved domains, namely, the C-terminal homeodomain (HD), the N-terminal MEINOX domain (containing KNOX1 and KNOX2) [28], and the ELK domain with nuclear localization signaling function located upstream of the HD [29]. The first *KNOX* gene identified in the plant kingdom was *KNOTTED1* (*KN1*), associated with leaf phenotypic difference in maize [30].

A series of studies have confirmed that *KNOX* genes in plants are involved in regulating leaf growth and development. A study of *FaKNOX1* obtained from cultivated strawberry (*Fragaria* spp. × *ananassa*) showed that its overexpression or repression caused abnormal leaf morphology. Overexpression of *FaKNOX1* led to leaf shrinkage, RNAi plants exhibited deeply serrated leaves and abnormally elongated petiolules, suggesting that appropriate *KNOX* gene expression level is necessary for maintaining normal leaf growth and development [31]. A study of double mutants found that the *KNOX1* gene in maize was directly involved in the GAs signaling pathway. Multiple biochemical analyses showed that the normal expression of the *KNOX1* gene inhibited the accumulation of active GAs, especially in immature leaves [32]. A study using a semi-dominant *KN1* mutant in maize showed that the accumulation of GA2ox1, which inactivates GAs, was positively correlated with the expression level of *KN1*. In addition, several genes encoding auxin response factor (ARF) are likely to be targeted by *KN1* [33]. In a study on *Hairy Sheath Frayed1* (*Hsf1*), a leaf mutant with abnormal leaf shape indicated that the *KNOX* gene may promote cytokinin (CK) accumulation; then, the CK signal was further transduced by Hsf1 to promote leaf sheath development [34]. In addition, several recent studies have shown that the *KNOX* gene is also involved in the regulation of biological processes such as rice grain size [35], abiotic stress tolerance of *Dendrobium huoshanense* [36] and mungbean seed dormancy [37].

The term “transcriptome” was first coined when studying the characteristics of yeast cells in different cell cycles [38]. Transcriptomics, an emerging discipline based on it, can not only analyze the expression of specific genes in cells, but also serve as an important method to interpret the rules of transcription regulation [39]. With the continuous upgrading of sequencing technology, transcriptomics can be combined with sequencing technology, which has given rise to a new technology, transcriptome sequencing (RNA-seq), which has the characteristics of high sensitivity, high resolution and wide application, especially for new transcripts and low-abundance transcripts. As such, it has attracted wide attention from researchers. The high sensitivity of RNA-seq makes it suitable for the identification of genes involved in molecular mechanisms related to plant growth and development, including genes involved in regulating maize LA, which provides an important theoretical basis for further study of the functions of related genes. Using RNA-seq, Johnston et al. compared transcripts in B73 and mutant *liguleless1-R* leaf primordia and identified a series of transcripts that were specifically upregulated at the leaf–sheath boundary and confirmed that these genes are involved in multiple phytohormone signaling pathways [40].

At present, the regulation of KNOX TFs on LA has been studied to some extent in maize and rice [41], but it is relatively rare compared with bHLH and other TFs, and the interaction between known *KNOX* gene family members still needs to be clarified in detail. In this study, a mutant with relatively small LA, *kn1*, was obtained by ethylmethane sulfonate (EMS) mutagenesis with maize inbred line B73 as the wild type (WT), and its various shapes were analyzed. Based on the results of bioinformatics analysis of *KN1* including *cis*-acting elements, we selected exogenous IAA and ABA to treat B73 and *kn1* to observe changes in LA. Combined with bioinformatics analysis including RNA-seq and exogenous phytohormone spraying experiments, we initially explored the molecular mechanism of *KN1* involved in regulating LA of maize under different conditions, which laid a theoretical foundation for further breeding maize ideal plant type with suitable LA.

## 2. Results

### 2.1. Identification and Basic Bioinformatics Analysis of KN1

According to the information of B73 reference genome RefGen_v4, *KN1* (*Zm00001d033859*) is located on chromosome 1, with a total DNA sequence length of 1682 bp and a coding sequence (CDS) length of 1080 bp, and the amino acid number of the end-product of gene expression is 359. DNA sequences of *KN1* in WT B73 and mutant *kn1* were extracted and sequenced, respectively, to identify the mutation sites. Sequencing results showed that in mutant *kn1*, the 853rd base in CDS of *KN1* gene changed from “C” to “T”, resulting in the codon at this site changing from CAG (glutamine) to TAG (termination codon) (Figure 1A). Mutation at this site shortened the protein product, ultimately resulting in phenotypic differences between the mutant and the WT. The amino acid sequence of KN1 was analyzed using blastp in the NCBI database, and MEGA 7.0 software was used to visualize homologous relationships between related amino acid sequences. The results showed that *KN1* had the highest homology with *KNOX* gene family members in *Sorghum bicolor* L. (Figure 1B). In order to initially understand the function and expression characteristics of *KN1* in the growth and development of maize leaves, the expression pattern of *KN1* in roots, stems, old leaves, young leaves and the pulvinus of the second leaf of B73 at V3 growth stage (when the third leaf is fully unfolded) was detected by qRT-PCR. Results showed that *KN1* had the highest expression level in stems and the pulvinus of the second leaf, followed by the old leaves and young leaves, and the lowest expression level in roots (Figure 1C). According to the search results in MaizeGDB (https://www.maizegdb.org/ (accessed on 20 March 2023)), the expression of *KN1* was involved in the growth and development of internode, ear primordium and female ears (Figure 1D). In addition, *cis*-acting elements in the 1500-bp regulatory region upstream of the *KN1* promoter (ATG) were analyzed. The results showed that the promoter region of *KN1* gene contains *cis*-acting elements in response to many phytohormones such as salicylic acid (SA), methyl jasmonate (MeJA), ABA, IAA and GAs. In addition, we identified *cis*-acting elements related to stress response, growth and development, and protein metabolism (Table 1). These results suggest that *KN1* may regulate the growth and development of maize leaves by participating in plant hormone signal transduction and provide a theoretical basis for the subsequent selection of ABA and IAA for exogenous phytohormone spraying experiments.

### 2.2. Identification of Phenotype and Grain Characteristics of Mutant kn1

The phenotypic characteristics of WT B73 and mutant *kn1* were analyzed, and the detailed results were as follows: Compared with WT B73, the PH and ear height of mutant *kn1* at maturity were significantly reduced by 18.93% and 13.89%, respectively. The investigation results and difference analysis of other field traits are shown in Table 2. In terms of LA and PH at maturity, the measurement results showed that the LA and PH of *kn1* were significantly smaller than those of WT (Figure 2A,B). In terms of grain traits (Figure 2C), there were significant differences between WT and *kn1*; the measurement results showed that compared with the WT, the ear diameter and 100-grain weight of *kn1* increased by 4.20% and 30.32%, respectively. On the contrary, the weight per ear, the length of ear, the number of ear rows and the number of grains per row of *kn1* decreased by 20.49%, 1.23%, 8.00% and 27.50%, respectively. The above measurements indicate that the mutation of *KN1* caused a variety of phenotypic differences.

### 2.3. Spraying of Exogenous Phytohormones and Determination of Optimal Concentration

In order to explore the regulation of KNOX TF on maize LA and the specific effects of exogenous phytohormones on maize LA, two phytohormones, ABA and IAA, were applied to exogenous treatment of wild type B73 and mutant *kn1* individuals, and LA was measured at V3 growth stage. The results showed that the LA of B73 was greater than that of *kn1* when treated with exogenous ABA at a concentration of 0 μM (Mock), which was consistent with the size of the results obtained from field data. With the exogenous ABA concentration gradually rising to 10 μM, the LA of B73 decreased first and then increased, while that of *kn1* increased gradually, and the LA of both reached the maximum value when the exogenous ABA concentration was 10 μM. When the concentration of exogenous ABA continued to increase from 10 μM, the LA values of B73 and *kn1* began to decrease (Figure 3A,B). When the concentration of exogenous IAA gradually increased from 0 to 100 μM, the LA of B73 and *kn1* showed a steady increase trend, and reached the maximum value when the concentration of IAA was 100 μM. When the concentration of exogenous IAA was higher than 100 μM, the LA of both began to decrease (Figure 3C,D). The above phytohormone spraying experiments showed that the optimal concentrations of exogenous ABA and IAA were 10 μM and 100 μM, respectively. Moreover, the increase in the LA of *kn1* treated with exogenous phytohormones was greater than that of B73, indicating that *kn1* exhibited greater sensitivity to exogenous phytohormones than B73.

### 2.4. Transcriptome Sequencing (RNA-seq) Statistics and Transcriptomic Profile of B73 and kn1

In order to obtain a comprehensive representation of the transcriptome of B73 and *kn1*, as well as the mechanism of response to exogenous phytohormones, Illumina high-throughput sequencing technology was applied to samples from B73 and *kn1*. Each B73 and *kn1* sample yielded more than 43 million and 20 million clean data, respectively. All samples contained more than 89% high-quality bases (Q30). The alignment results showed that at least 94% of clean reads in each sample could be mapped to the B73 reference genome. The lowest proportion of reads that were uniquely mapped to the reference genome and the highest proportion of reads that were multiple-mapped to the reference genome were 94.37% and 4.08%, respectively (Appendix A).

Differentially expressed genes (DEGs) in all samples were screened according to thresholds set in the DESeq R package (v1.18.0) (|log_2_Foldchange| > 1, q < 0.05) (Figure 4 and Appendix A). A total of 1105 DEGs were obtained in the Mock group without phytohormone treatment, with 783 upregulated and 322 downregulated DEGs (Appendix A). A total of 1408 DEGs were obtained in the group of B73 and *kn1* treated with 10 μM exogenous ABA, among which 782 upregulated and 626 downregulated DEGs were obtained, respectively (Appendix A). A total of 2074 DEGs were obtained in the group of B73 and *kn1* treated with 100 μM exogenous IAA, among which 1569 upregulated and 505 downregulated DEGs were obtained, respectively (Appendix A). There were 885 (552 upregulated and 333 downregulated) and 237 (87 upregulated and 150 downregulated) DEGs of B73 (Appendix A) and *kn1* (Appendix A) before and after 10 μM exogenous ABA treatment, respectively. There were 345 (114 upregulated and 231 downregulated) and 1095 (738 upregulated and 357 downregulated) DEGs of B73 (Appendix A) and *kn1* (Appendix A) before and after 100 μM exogenous IAA treatment, respectively.

### 2.5. Gene Ontology (GO) and Kyoto Encyclopedia of Genes and Genomes (KEGG) Enrichment Analysis of LA Mutant kn1

GO and KEGG enrichment analysis was performed on all DEGs in LA mutant *kn1* and WT B73, and the results were as follows: Functional annotations of all GO entries were divided into three categories: biological processes (BP), cell components (CC) and molecular functions (MF). Totals of 137 DEGs (101 upregulated and 36 downregulated), 183 DEGs (147 upregulated and 36 downregulated) and 451 DEGs (328 upregulated and 123 downregulated) were significantly enriched in BP-, CC- and MF-related GO terms, respectively (q < 0.05, the same as below). Compared with B73, BP-related DEGs in *kn1* were mainly enriched in reproductive development, lipid metabolism, embryonic development and tricarboxylic acid cycle. DEGs related to CC were mainly concentrated in the nucleus, cellular membrane components, ribosomes, mitochondrial outer membrane. DEGs related to MF were mainly concentrated in nucleic acid binding, exonuclease activity, hydrolase activity, protein serine/threonine kinase activity, RNA binding, ATP binding (Figure 5A). In order to further understand the specific biological functions of related DEGs and potentially related signaling pathways, the DEGs produced by *kn1* were mapped to KOBAS software (v2.1.1) for annotation. The results showed that the upregulated DEGs in *kn1* were mainly involved in phytohormone signal transduction, plant MAPK signaling pathway, tryptophan metabolism and plant pathogen interaction, while the downregulated DEGs were mainly involved in carbon metabolism, DNA replication, nucleotide excision and repair, and benzoxazinoid biosynthesis (Figure 6A). The above results showed that the expression of genes related to phytohormone signal transduction, partial amino acid biosynthesis and metabolism increased in the mutants. Gene expression levels related to genetic information transmission, DNA replication and nucleotide excision repair were inhibited.

### 2.6. GO and KEGG Enrichment Analysis of kn1 and B73 Treated with Exogenous ABA and IAA

B73 and *kn1* were treated with optimal concentrations of ABA solution and RNA-seq was performed. The results showed that compared with B73, *kn1* produced 24 (5 upregulated and 19 downregulated), 37 (10 upregulated and 27 downregulated) and 117 (46 upregulated and 71 downregulated) differentially expressed genes related to BP, CC and MF, respectively. DEGs in BP category were significantly related to cellular processes, metabolic processes and biological regulation. All DEGs in the CC category were significantly enriched in related items of cellular anatomical entity, intracellular and protein-containing complex. DEGs in the MF category were mainly concentrated in items related to catalytic activity and binding (Figure 5B). The results of KEGG enrichment analysis showed that compared with B73 treated with ABA, DEGs in *kn1* treated with ABA were significantly enriched in flavonoid and flavonol biosynthesis. These results indicated that exogenous ABA may be involved in the regulation of LA by affecting phytohormone signal transduction and secondary metabolite synthesis (Figure 6B).

RNA-seq and enrichment analysis were performed on B73 and *kn1* treated with an optimal concentration of exogenous IAA solution. The results showed that compared with B73, *kn1* produced 139, 222 and 497 DEGs in BP, CC and MF, respectively. Similar to that, after ABA treatment, BP-related DEGs produced by *kn1* after an optimal concentration of exogenous IAA was mainly enriched in cell process, metabolic process and biological regulation entries. All CC DEGs were significantly enriched in cellular anatomical entity, intracellular and protein-containing complex. DEGs related to MF were mainly significantly enriched in the items of catalytic activity and binding (Figure 5C). The subsequent KEGG enrichment analysis results showed that compared with B73, the upregulated DEGs generated by *kn1* treated with exogenous IAA were mainly enriched in phytohormone signal transduction, carotenide biosynthesis and tryptophan metabolism. Downregulated DEGs were mainly enriched in flavonoid biosynthesis, phenylpropane biosynthesis, glycerol metabolism, MAPK signaling pathway and fatty acid biosynthesis (Figure 6C). The above enrichment analysis results showed that exogenous IAA significantly affected the synthesis of various secondary metabolites and phytohormone signal transduction in plants, which may be the way that exogenous IAA participated in the regulation of LA.

### 2.7. qRT-PCR Verification of Differentially Expressed Genes

Five common DEGs were randomly selected from the RNA-seq data of the three comparison groups of B73-Mock vs. *kn1*-Mock, B73-ABA vs. *kn1*-ABA and B73-IAA vs. *kn1*-IAA, and their expression patterns were verified by qRT-PCR. The verification results showed that the expression patterns and trends of the five selected DEGs in all comparison groups were the same as those in the RNA-seq data (Figure 7), which proved the reliability of the RNA-seq data in this study.

### 2.8. Identification of Potential Hub Genes Involved in Regulation of LA via Weighted Gene Co-Expression Network Analysis (WGCNA)

To explore the correlation between DEGs obtained from RNA-seq data and LA phenotypic data, DEGs meeting the criteria of “FPKM ≥ 10” were screened from the RNA-seq data for WGCNA analysis to mine hub genes. The analysis results showed that all the DEGs were clustered into four different modules (Figure 8A,B), among which the genes in the gray module did not have any characteristic expression patterns, so they were not clustered into the other three characteristic expression modules. The relationship between each co-expression module and LA was analyzed, and the results showed that among the three co-expression modules, the blue module showed the highest correlation with the LA of maize (Figure 8C). The expression profile of genes in the blue co-expression module is shown in the heatmap (Figure 8D). In order to further explore the relationship between the genes in the blue module and LA in maize and to mine the genes related to the regulation of LA, the STRING database was used to construct the protein–protein interaction (PPI) network for the genes in the blue module, and the confidence parameter was set as “comprehensive score > 0.4”. Co-expression network analysis showed that seven genes located in the center of the network were identified in the blue module, which were called hub genes (Figure 9). In order to further explore the biological functions of hub genes related to LA in maize, GO enrichment and KEGG pathway enrichment analysis were performed. GO enrichment analysis showed that the seven hub genes in the blue module were mainly involved in the plasma membrane, nucleus, cellular membrane components, cell wall biogenesis and cellulose biosynthesis. KEGG pathway enrichment analysis showed that the key genes in the blue module were mainly involved in plant–pathogen interaction (Appendix A).

## 3. Discussion

LA is an important factor affecting maize yield. Compact LA is the key phenotype involved in forming the ideal maize plant architecture. At present, the genes involved in LA formation can be divided into two categories according to whether they are involved in phytohormone signaling pathways. Genes that have not been confirmed to be related to phytohormone signaling mainly include *LIGULELESS1* (*LG1*), *LG2*, *LG3*, *LG4* and *LIGULELESS NARROW* (*LGN*). *LGN* encodes serine/threonine protein kinase [42], the coding products of *LG3* and *LG4* are both KNOX TFs [43], *LG1* and *LG2* encode SPL [44] and basic leucine zipper (bZIP) [45] TFs, respectively. As mentioned above, KN1, the first KNOX TF identified in maize, is also involved in regulating LA by adjusting leaf morphology [27]. A series of studies have shown that the above genes are specifically expressed in ligule, auricle and other tissues that constitute maize leaf organs. Mutations at different sites and degrees in these genes lead to abnormal development of ligule and auricle, resulting in significant reduction in LA. Combined with the sequencing results of B73 and mutant *kn1* and the existing conclusions, it can be seen that the reduction in LA of mutant *kn1* is caused by the loss of function of *KN1*. The latter contains genes involved in a variety of phytohormone signaling pathways including IAA, BRs, GAs, SLs, etc. Among the known phytohormones related to LA in maize and rice, except GAs, which have a double effect on LA according to its source, the other phytohormones are involved in the regulation of LA with a stable effect. A study using transgenic rice and *Arabidopsis thaliana* showed that high expression of *OsBCL1* and *OsBCL2* in the pulvinus of rice leaf significantly promoted GA_3_ biosynthesis, increasing LA via promoting cell elongation rather than cell division. *OsBCL1* and *OsBCL2* are homologs of *OsBC1*, which encodes a bHLH transcription factor [20]. An earlier study on the relationship between GAs and BRs in rice found that exogenous GAs significantly reduce LA. A subsequent series of analyses using multiple types of BRs metabolic mutants found that exogenous GAs inhibited the expression activities of *D2* and *D11* associated with BRs biosynthesis. In addition, exogenous GAs significantly inhibited BZR1, a positive regulator in the BRs signal transduction pathway, while activating GSK2, a key regulator that negatively regulated the BRs signaling pathway, suggesting the negative regulatory effect of exogenous GAs on LA and the complexity of its regulatory function [21]. A series of earlier studies demonstrated that the *KNOX1* gene in maize interacts with signaling pathways of a variety of phytohormones, including GAs and IAA. A study combining double mutant analysis and biochemical analysis demonstrated that the expression of *KNOX1* in maize hindered the normal biological function of GAs. A series of biochemical analyses, including the electrophoretic mobility shift assay (EMSA), demonstrated that the encoding product of *KNOX1* binds to a *cis*-acting element with two TGAC motifs in the first promoter of *GA2ox1*, activating transcription of *GA2ox1* to inhibit the accumulation of active GAs [32]. Studies on *KN1* semi-dominant mutants in maize showed that the expression level of *KNOX1* was significantly positively correlated with the accumulation of GA2ox1, which was consistent with previous findings. In addition, a series of analyses, including chromatin immunoprecipitation sequencing (ChIP-seq) and RNA-seq, have shown that KNOX1 protein can bind to genes encoding important members of the IAA signaling pathway such as *ARF* and *AUX/IAA*. These results indicate that *KNOX1* may be involved in the regulation of the IAA signaling pathway by targeting and regulating the expression activity of related genes [33].

IAA and ABA are extremely important phytohormones involved in a wide range of life activities, including growth and development and stress response [46,47]. Previous studies on IAA regulation of LA mainly focus on rice, and most of the known LA-related genes are involved in IAA biosynthesis and signal transduction. The encoding product of *OsGH3-1* is an enzyme that converts IAA in the IAA signal transduction pathway. Overexpression of *OsGH3-1* in rice inactivates IAA by binding amino acids, reducing the level of free IAA in plants and significantly increasing LA [14]. IAA can also induce the expression of non-coding RNA represented by miR393, which regulates leaf inclination by inhibiting the expression level of its target gene *OsTIR1* in rice, thus determining the formation of LA in rice [15]. In addition, IAA can also participate in the regulation of LA by intervening in other phytohormone signal transduction. For example, some ARF regulatory elements in the IAA signaling pathway in rice have been shown to increase the expression levels of genes such as *OsBRI1* and *OsGH3s*, thus increasing the value of LA [24,48]. Compared with IAA, there are relatively few studies on the involvement of ABA in LA formation. Early studies on *Arabidopsis thaliana* have shown that WRKY family TFs containing ABA-responsive elements are involved in plant response to ABA as positive regulatory elements [49]. Phenotypic investigation of *OsWRKY53*-overexpressed rice plants and *oswrky53* mutants showed that overexpression of *OsWRKY53* significantly increased LA and grain size; correspondingly, the mutant had reduced LA and grain size. A series of subsequent studies showed that the expression level of *OsWRKY53* was positively correlated with the sensitivity of plants to BRs, suggesting that *OsWRKY53* plays a positive regulatory role in the BRs signaling pathway [50]. A recent study on the regulation of maize LA showed that the gene controlling maize LA, *ZmCLA4*, which was cloned in QTL *qLA4-1* [51], was involved in the regulation of maize LA by encoding a protein as a transcriptional repressor. A series of subsequent analyses, including DNA affinity purification sequencing (DAP-seq), demonstrated that ZmCLA4 inhibited the expression activity of a series of genes by binding to promoters containing motifs such as ERF, and the target genes regulated by ZmCLA4 were involved in the signal transduction process of various phytohormones, such as IAA (*ZmARF22* and *ZmIAA26*), BRs (*ZmBZR3*), ABA (two *ZmWRKY*-like genes), jasmonic acid (JA) (*ZmCYP75B1* and *ZmCYP93D1*) and ethylene (*ZmABI3*) [52]. Li et al. produced corresponding mutants from three different japonica rice cultivars. A series of analyses, including an exogenous phytohormone spraying experiment, RNA-seq and yeast two-hybridization, demonstrated that ABA interferes with the positive regulation of BRs by targeting the expression activity of genes involved in BRs synthesis (*D11*) and BRs signal transduction (*GSK2* and *DLT*), suggesting the negative role of ABA in the regulation of LA and the complexity of the interaction between ABA and other phytohormones [26]. These findings provide new insights into the interaction between different phytohormones in regulating LA in maize.

As mentioned earlier, phytohormones are one of the core factors influencing LA formation. By analyzing DEGs produced by B73 and *kn1* under different exogenous phytohormones, we found that the genes encoding Aux/IAA, GA 3-oxidase (GA3ox), flavone synthetase (FNS), dihydroflavonol-4-reductase (DFR), xyloglucan endotransglucosylase/hydrolase (XTH), polygalacturonase inhibitor (PGI) showed different differentially expressed trends in B73 and *kn1* (Appendix A). Aux/IAA is an important regulator of IAA signal transduction and usually forms a dimer with ARF, which has the function of TF, to inhibit the expression of related downstream genes [53]. GA3ox is a key member of GAs biosynthesis in plants, which catalyzes the production of GA_1_ and GA_4_ to increase the content of bioactive GAs in plants [54]. FNS and FDR are key enzymes involved in flavonoid biosynthesis. Studies on Arabidopsis mutants with low flavonoid content have shown that auxin transport levels are inhibited by flavonoids [55]. XTH and PGI participate in the cell wall remodeling process by catalyzing glycosyl transfer [56] and inhibiting the activity of polygalacturonase (PG) [57], respectively, and then participate in the control of plant morphology by affecting cell elongation. According to the expression of the above related genes in B73 and *kn1* after treatment with different exogenous phytohormones, we summarized the possible molecular model of exogenous phytohormones involved in the regulation of LA (Figure 10). After exogenous ABA treatment, the IAA polar transport in B73 was blocked, the expression activity of downstream response gene *XTH* in the BRs signaling pathway was enhanced, and the inhibition of PG was alleviated. (Figure 10A). Only enhanced PG activity was identified in *kn1* after exogenous ABA treatment (Figure 10C). Therefore, we hypothesized that the higher flavonoid content in B73 inhibited IAA polar transport, thus blocking the IAA-mediated cell wall softening process, and the gene expression involved in flavonoid biosynthesis changed more significantly (log_2_Foldchange > 2), which cancelled out the positive regulatory effect of BRs signal on LA, resulting in a larger increase in the LA of *kn1* after exogenous ABA treatment. After exogenous IAA treatment, flavonoid biosynthesis was hindered in B73 and *kn1*. Two XTH and Aux/IAA coding genes were upregulated in *kn1* (Figure 10D), and the expression of the former was higher (log_2_Foldchange > 2), while only one GA3ox coding gene was upregulated in B73 (Figure 10B). Accordingly, we speculated that the GAs signal enhancement effect caused by exogenous IAA in B73 was partially cancelled, and the IAA and BRs signals in mutant *kn1* were more strongly inhibited and promoted, respectively, and thus showed a larger increase in LA than B73. These results suggest that BRs plays an important role in the regulation of LA and that exogenous IAA and ABA may be involved in *KN1* regulation of LA. The detailed function and related signal network of *KN1* in the regulation of LA still need to be further studied.

## 4. Materials and Methods

### 4.1. Plant Materials and Planting Circumstances

Maize mutant *kn1* with relatively reduced LA was provided by the maize EMS mutant library of Qilu Normal University, and WT cultivar B73 with slightly larger LA was provided by the Maize Breeding Research Team of the Agricultural College of Gansu Agricultural University. Female panicles of B73 were pollinated by EMS-treated B73 pollens to obtain the M_1_ mutant. Afterward, M_1_ and B73 underwent two generations of backcross and one generation of selfing. The Sanger method was used to observe the mutant phenotype of each generation and remove the mutant background. The alternate planting technique was used to plant mutant and WT materials in the experimental field of Gansu Agricultural University. The row width, inter-row spacing and plant spacing were set at 70 cm, 40 cm and 25 cm, respectively.

### 4.2. Analysis of kn1 Mutant Phenotype

After pollination, 3 plants with robust and similar growth were selected from the mutant and WT materials to measure the field phenotype data (plant height, ear height, chlorophyll content, stem diameter, leaf length, leaf width, leaf direction value and leaf area size). A total of 10 mutant and 10 WT plants with good and similar growth were selected to measure the ear LA, the LA above the ear and the LA below the ear. Mature ears were harvested at the end of September of the same year. After full drying to safe moisture content, thresh treatment was carried out and related grain characteristics data (ear weight, ear length, ear diameter, 100-kernel weight, number of ear rows, ear rows, cob weight and cob diameter) were measured. The following formula was used to calculate leaf area size (LAS) and leaf direction value (LDV):LAS = LL × LW × 0.75
LDV = ∑(90 − θi) × (LS/LL)/Q
where LL, LW, θi, LS and Q denote the leaf length, leaf width, LA value, leaf spacing and quantity of leaves above the ear, respectively [58].

### 4.3. Bioinformatics-Related Analysis of KN1

NCBI database was used to search the nucleotide sequence of *KN1* and the amino acid sequence of its expression products. Sequence and sequence alignment of *KN1* were analyzed via DNAMAN 6.0 software. ClustalW (v2.0.11) and MEGA 7.0 were used to perform multiple alignment of amino acid sequences and form multiple alignment results generated by ClustalW. A phylogenetic tree was generated using the maximum likelihood method with pairwise deletion; the bootstrap value was set to 1000. The 1500 bp sequence upstream of the translation start codons of *KN1* was downloaded from the maizeGDB database and then submitted to the PlantCARE database [59] for analysis of the promoter region of *KN1*.

### 4.4. Identification of Homozygous Mutant kn1

Young leaves of WT B73 and mutant *kn1* at V3 growth stage were collected from the field, and their DNA was extracted. Primers were designed using Primer Premier 5.0 software based on the 100 bp interval length of the upstream and downstream mutation sites. The forward and reverse primer sequences were 5′-ACTAACATTGGGTAGCTGGCT-3′ and 5′-GCTCCCTGAAGTTGATGCAC-3′, respectively. The mutation sites were detected by polymerase chain reaction (PCR) amplification and product sequencing. The PCR reaction system (20 μL) included 1 μL DNA template, 1 μL each of the forward and reverse primers, 7 μL ddH_2_O, and 10 μL 2× SanTaq PCR Mix (Sangong Bioengineering Co., Ltd. Shanghai, China). The products tested by 1% agarose gel electrophoresis were sent to Sangong Bioengineering Co., Ltd. (Shanghai, China) for sequencing. DNAMAN 6.0 software was used to compare the sequencing results of the PCR products of the mutant to determine the mutation site.

### 4.5. Expression Pattern Analysis of KN1

The root, stem, pulvinus of the second leaf, old leaf (second leaf) and new leaf (fourth leaf) of wild type B73 at V3 growth stage were collected, frozen and stored in liquid nitrogen. Total RNA was extracted, and its concentration and integrity were determined using a SteadyPure plant RNA extraction kit and spectrophotometer, respectively. cDNA was synthesized and purified with a M-MLV (Moloney Murine Leukemia Virus) reverse transcription kit and gDNA Clean Reagent, respectively. Total RNA and cDNA were stored in a −80 °C ultralow-temperature refrigerator and −20 °C refrigerator, respectively, for subsequent experimental analysis. Reverse transcription conditions were set as “37 °C for 15 min, 85 °C for 5 s”.

The SYBR Green Pro Taq HS premixed qRT-PCR kit (Sangong Bioengineering Co., Ltd. Shanghai, China) was used for qRT-PCR analysis. The reaction system (20 µL) consists of 2 x SYBR Green Pro Taq HS Premix (Sangong Bioengineering Co., Ltd. Shanghai, China) (10 µL), cDNA (1 µL), positive and negative primers (0.4 µL each), RNase-free water (8.2 µL). The reaction procedure consists of an initial denaturation step (95 °C for 3 min) and a subsequent cycle step (20 s at 95 °C, 20 s at 59 °C and 25 s at 72 °C). The Roche LightCycler96 PCR instrument (Roche, Switzerland) was used to complete the amplification procedure. The primers required for validation of expression patterns are shown in Appendix A. The relative expression levels of genes were normalized with Actin and calculated via the 2^−ΔΔCt^ method [60].

### 4.6. Determination of the Optimal Concentration of Exogenous Phytohormones

Seeds of WT B73 and mutant *kn1* with complete, full and consistent grain shape were selected and planted in pots mixed with vermiculite after soaking for 24 h. Six seeds were seeded in each pot, and three biological replicates were set for B73 and *kn1*. All plants were placed in light incubator (12 h light/12 h dark, 28 °C, 65% RH) for growth and seedling traits were measured. In this assay, B73 and *kn1* seedling individuals were treated with ABA and IAA solution, respectively. The former was set at four concentrations (0 μM, 1 μM, 10 μM and 50 μM); the latter was set at five concentrations (0 μM, 1 μM, 10 μM, 100 μM and 500 μM). After the seedlings grew to V1 (when the first leaf is fully unfolded) growth stage, they were treated according to the above concentration. A total of 50 mL of phytohormone solution was poured every 2 days. At the same time, the same concentration of phytohormone solution was sprayed on their leaves. The control group (Mock) was set to treat B73 and *kn1* with the same volume of distilled water as the phytohormone solution. At V3 growth stage, the LA of the second leaf was measured to screen out the best concentration of exogenous phytohormone.

### 4.7. Transcriptomic Profiles and Data Analysis

The pulvinus of the second leaf of B73 and *kn1* individuals in V3 growth stage from the control group and hormone-treated groups was collected and frozen in liquid nitrogen for subsequent RNA-seq analysis. Three biological replicates were set for each group, a total of 18 samples. Total RNA was extracted from each plant sample using TRIzol reagent, then the purity and concentration was measured using NanoDrop 2000 (Thermo Fisher Scientific, Wilmington, DE, USA). Finally, an RNA Nano 6000 Assay Kit was used to check the integrity of the RNA via an Agilent Bioanalyzer 2100 system (Agilent, Santa Clara, CA, USA). Sequencing was performed using the Illumina Hiseq TM2500 platform (Biomarker Technologies, Beijing, China). CASAVA Base Calling was used to convert the raw image data files obtained during sequencing into raw sequencing data. The sequencing results were saved as fastq format files that record sequencing information and quality for subsequent analysis.

The quality control and mapping of sequencing data were completed by fastp (v0.20.0) [61] and Hisat2 (v2.0.4) [62], respectively. Fragments per kilobase of transcript sequence per million (FPKM) values, calculated based on the length of the fragments and read counts mapped to corresponding fragments [63], were used as an indicator to measure the level of gene expression. The DESeq2 R package (v1.30.1) [64] was used to perform differential expression analysis between samples with different phenotypes. Genes meeting the criteria of |log_2_Foldchange| > 1 and q < 0.05 were defined as differentially expressed genes (DEGs). The q-value represents the *p*-value corrected with false discovery rate [65]. Enrichment analysis of DEGs in all samples was performed by clusterProfiler R package (v4.4.4) and KOBAS (v2.1.1). The original sequencing dataset generated in this study has been uploaded to the Sequence Read Archive (SRA) database with the accession number SRP488570.

### 4.8. Reliability Verification of RNA-seq via qRT-PCR

DEGs were randomly selected from the RNA-seq data for qRT-PCR validation. Total RNA extraction, cDNA synthesis and reaction procedures were performed as described above. The primers required for reliability verification of RNA-seq data are shown in Appendix A.

### 4.9. Weighted Gene Co-Expression Network Analysis

The WGCNA package (v1.47) in R [66] was used to perform co-expression analysis based on RNA-seq data. DEGs that met the criteria of “FPKM ≥ 10” in the sequencing data were used for co-expression network analysis to explore molecular mechanisms that might be involved in regulating LA. The relevant parameters were set as follows: soft threshold β = 18. The threshold value is the value when the fitting curve is located at 0.9. The topological overlap measure (TOM) value was not set. The minimum value of eigengene-based connectivity (kME) was set to 0.7. The minimum number of genes contained in each module is 30. The maximum number of genes that can be displayed by the co-expression network is 150. The potential hub genes in the co-expression network were analyzed by Cytoscape software (v3.7.1) [67].

## 5. Conclusions

In this study, WT inbred line B73 and mutant *kn1* generated via EMS treatment were used as experimental materials. It was identified that the “C-to-T” mutation at the 1098th base of the *KN1* coding sequence terminated the transcription process of *KN1* in advance, resulting in a truncated protein, which eventually led to a series of phenotypic differences, including PH and LA. Seedlings of B73 and *kn1* were treated with exogenous phytohormones to screen the optimal concentrations of phytohormones used (10 μM ABA and 100 μM IAA). RNA-seq was performed to analyze the changes in transcriptional levels caused by mutations in *KN1*. RNA-seq results showed that genes involved in biological functions and pathways such as the MAPK signaling pathway, serine/threonine protein kinase, cellulose biosynthesis and cell wall formation were differentially expressed in B73 and *kn1*, suggesting that *KN1* may be involved in the cascade reaction pathway of MAPK and serine/threonine protein kinase, and may be related to biological processes such as cellulose biosynthesis and cell wall formation. Seven potential key genes involved in the regulation of LA were identified in WGCNA analysis of B73 and *kn1*, which were significantly related to cell membrane components, cellulose biosynthesis and cell wall formation (Appendix A). Comparative analysis of DEGs showed that the function of *KN1* was probably related to the IAA, BRs and GAs signal transduction pathways; flavonoid biosynthesis; and PG- and XTH-mediated cell wall remodeling. This study suggests that *KN1* may be related to cell wall-mediated changes in LA and provides new insights into the formation of LA in maize.

## Figures and Tables

**Figure 1 ijms-25-03180-f001:**
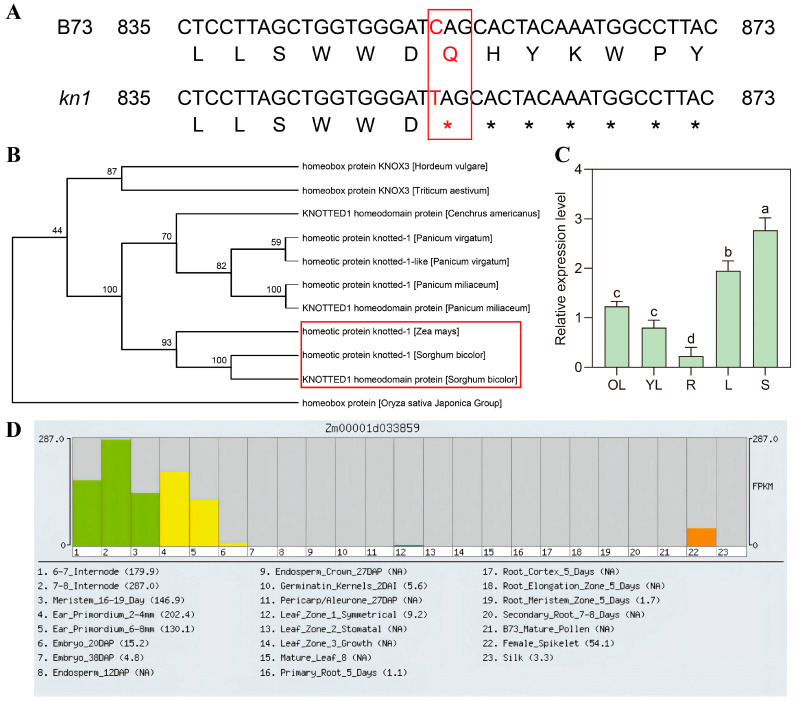
Basic bioinformatics analysis of *KN1* gene in maize. (**A**) Partial sequence alignment analysis of coding sequence of the *KN1* gene in wild-type B73 and mutant *kn1*. * indicates sites where the corresponding amino acid does not exist. The red letters in the DNA sequence indicate the site of the mutation induced by EMS. The red letters in the amino acid sequence indicate amino acid residues that have changed as a result of mutations. The range marked in the red box indicates the locations in the DNA sequence where mutation occurs and the location in the amino acid sequence where mutations affect amino acid type changes. (**B**) Phylogenetic tree constructed based on the *KN1* gene in maize. The red box indicates the position of KN1 in maize and the KN1 protein with the highest similarity to it in the phylogenetic tree. (**C**) Relative expression levels of *KN1* gene in various tissues. Different lowercase letters represent a statistically significant difference with a confidence level of “*p* < 0.05” (one-way analysis of variance, Tukey’s method). (**D**) *KN1* gene relative expression dataset retrieved from MaizeGDB website. Note: OL–old leaves; YL–young leaves; R–roots; L–the pulvinus of the second leaf at the V3 growth stage; S–stems.

**Figure 2 ijms-25-03180-f002:**
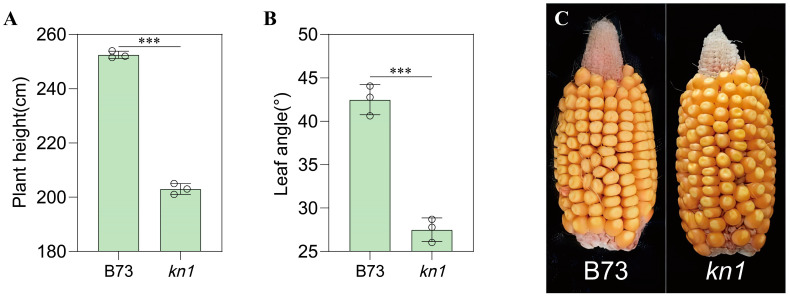
Analysis of major phenotypic traits in B73 and *kn1*. (**A**) Results of plant height measurements for B73 and *kn1* (n = 3 biological replicates). (**B**) Results of LA measurements for B73 and *kn1* (n = 3 biological replicates). (**C**) Mature ear phenotypes of B73 and *kn1*. Note: Unpaired two-tailed Student’s *t*-tests was used to detect the significance of differences between the two groups. *** indicates *p* < 0.001. LA–leaf angle. Data are expressed as mean ± SD.

**Figure 3 ijms-25-03180-f003:**
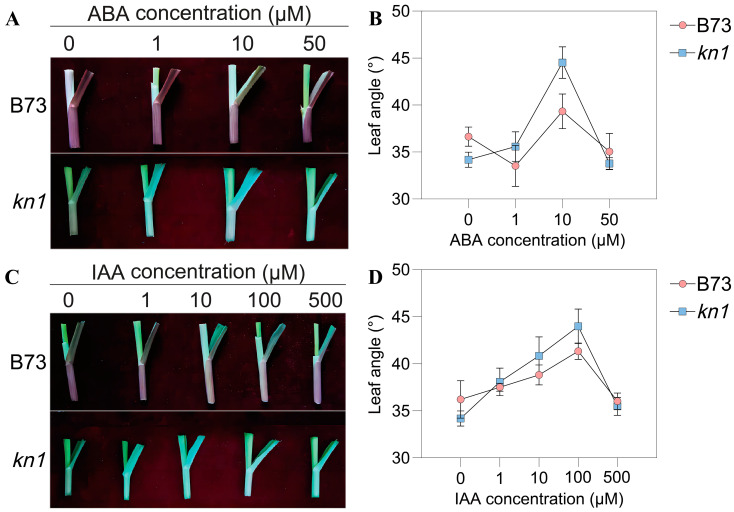
Effect of exogenous spraying of ABA and IAA on LA in B73 and *kn1*. (**A**) Phenotypic changes in B73 and *kn1* treated with different concentrations of ABA. (**B**) LA measurements of B73 and *kn1* treated with different concentrations of ABA (n = 3 biological replicates). (**C**) Phenotypic changes in B73 and *kn1* treated with different concentrations of IAA. (**D**) LA measurements of B73 and *kn1* treated with different concentrations of IAA (n = 3 biological replicates). Note: LA–leaf angle. Data are expressed as mean ± SD.

**Figure 4 ijms-25-03180-f004:**
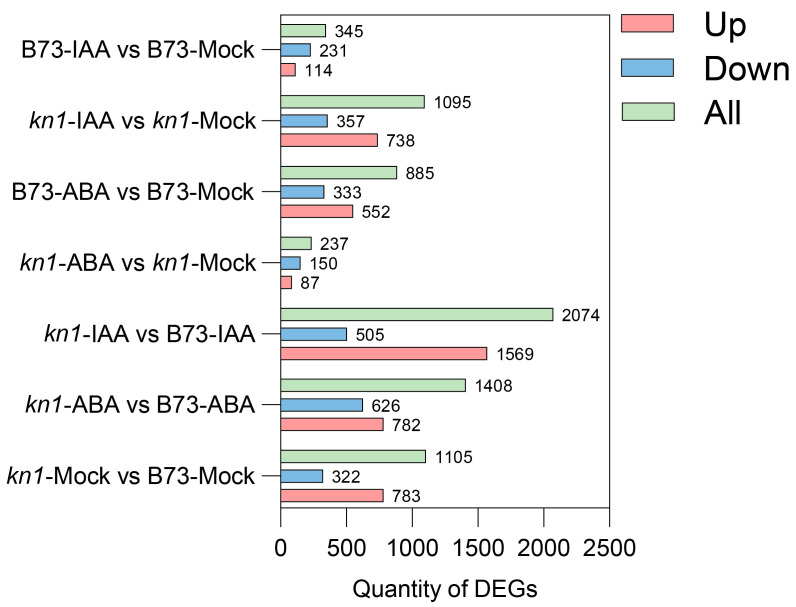
Summary of the quantity of DEGs identified in each comparison. Note: DEGs represents differentially expressed genes. Mock represents exogenous ABA or IAA at a concentration of 0 µM, namely, the control group.

**Figure 5 ijms-25-03180-f005:**
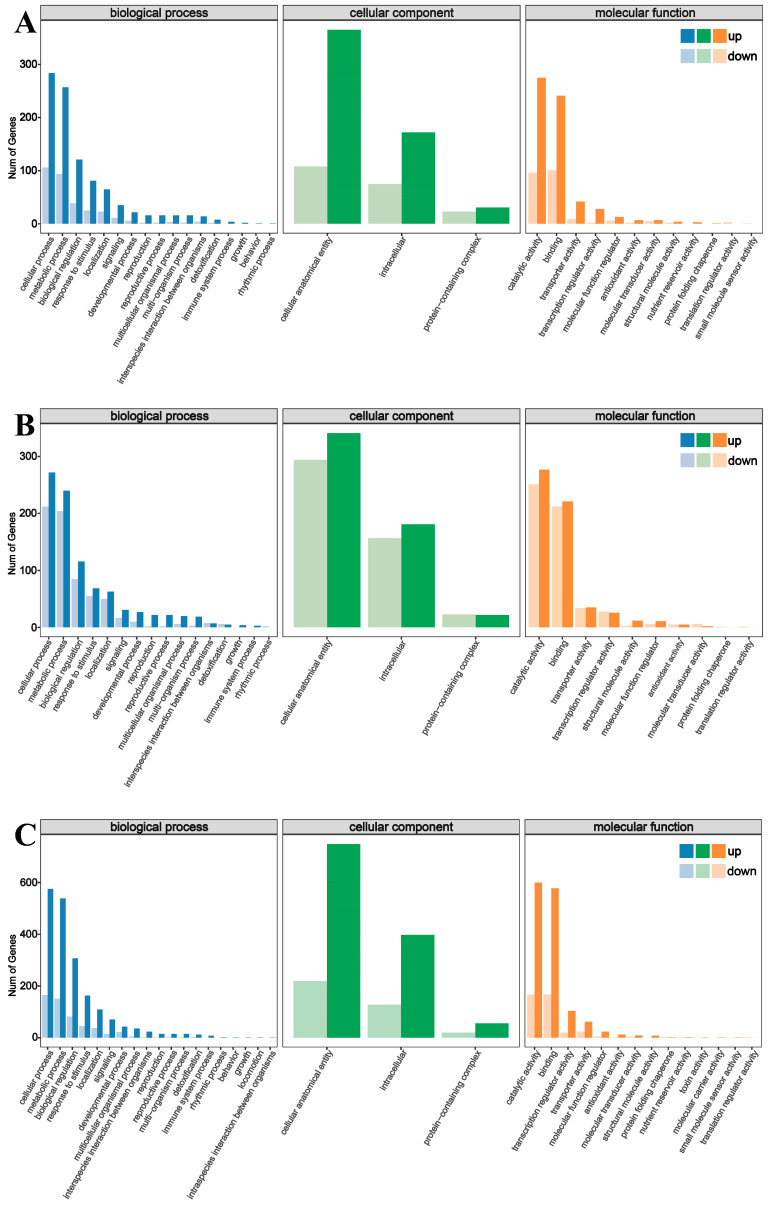
GO enrichment analysis of DEGs produced by B73 and *kn1* under different treatment conditions. (**A**) GO enrichment analysis of B73-Mock vs. *kn1*-Mock; (**B**) GO enrichment analysis of B73-ABA vs. *kn1*-ABA; (**C**) GO enrichment analysis of B73-IAA vs. *kn1*-IAA.

**Figure 6 ijms-25-03180-f006:**
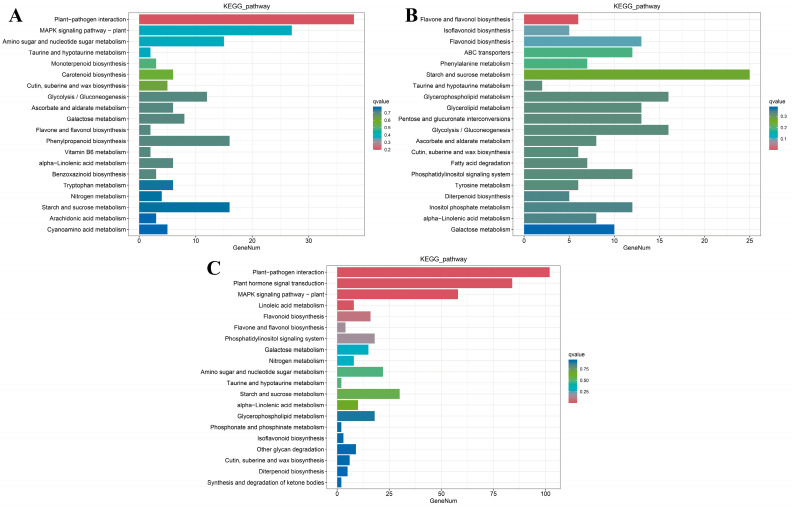
KEGG pathway enrichment analysis of DEGs produced by B73 and *kn1* under different treatment conditions. (**A**) KEGG pathway enrichment analysis of B73-Mock vs. *kn1*-Mock; (**B**) KEGG pathway enrichment analysis of B73-ABA vs. *kn1*-ABA; (**C**) KEGG pathway enrichment analysis of B73-IAA vs. *kn1*-IAA.

**Figure 7 ijms-25-03180-f007:**
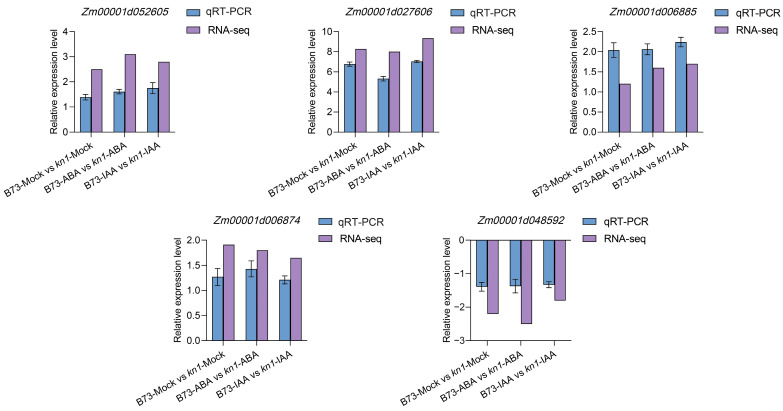
qRT−PCR validation analysis of five randomly selected DEGs from transcriptome sequencing data.

**Figure 8 ijms-25-03180-f008:**
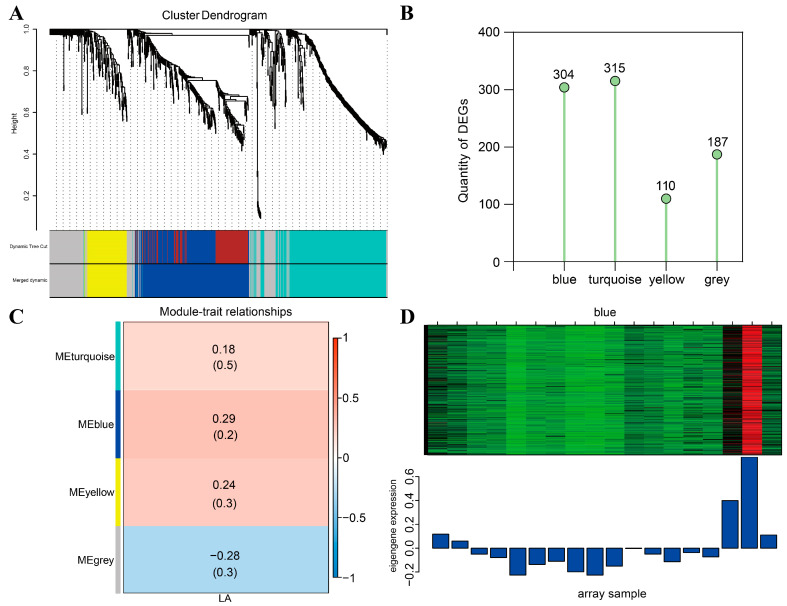
Weighted gene co-expression network analysis based on the difference in LA between B73 and *kn1* under Mock, exogenous ABA and IAA treatments. (**A**) Hierarchical clustering dendrogram of gene co-expression modules. (**B**) Statistical overview of the quantity of genes contained in each module. (**C**) Heatmap of correlation between gene co-expression modules and LA. Values in each cell represent the coefficient value of correlation (above) and *p*-values (below) of the co-expression module–LA association. (**D**) Heatmap of genes and eigengene expression in blue module. Note: LA–leaf angle.

**Figure 9 ijms-25-03180-f009:**
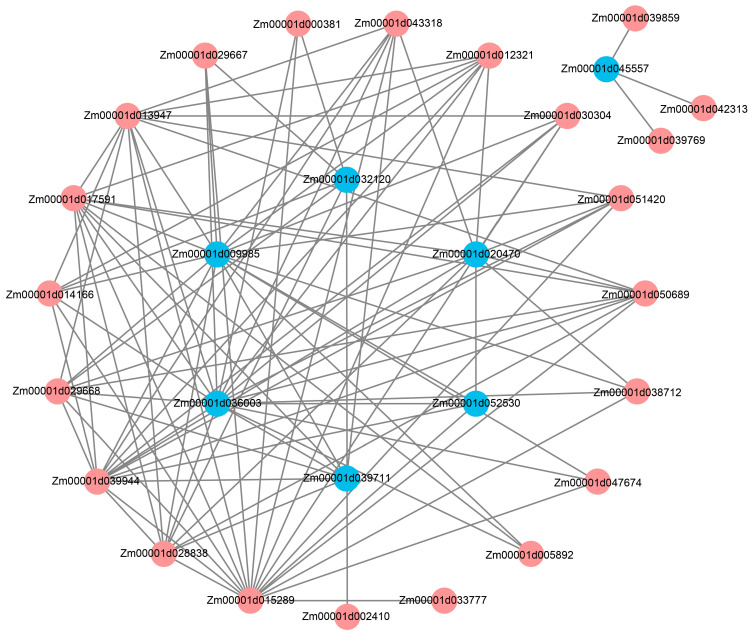
Co-expression regulatory network of genes in blue module. The blue nodes in the figure represent hub genes in the regulatory network. The remaining red nodes indicate genes in the co-expression network that do not have key regulatory roles.

**Figure 10 ijms-25-03180-f010:**
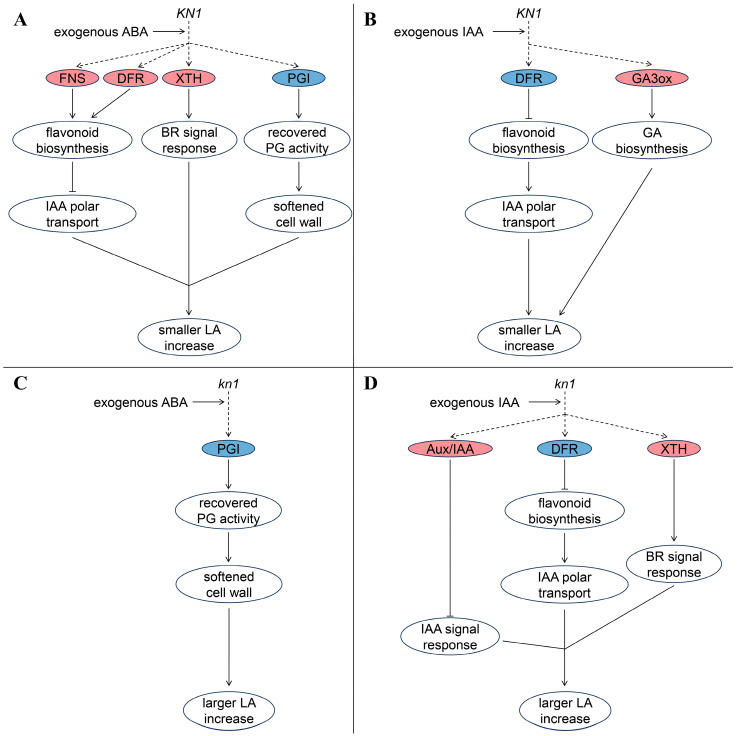
A possible molecular model of *KN1* regulating LA with exogenous phytohormones. (**A**) Molecular model of WT *KN1* regulating LA under exogenous ABA treatment. (**B**) Molecular model of WT *KN1* regulating LA under exogenous IAA treatment. (**C**) Molecular model of mutant *kn1* regulating LA under exogenous ABA treatment. (**D**) Molecular model of mutant *kn1* regulating LA under exogenous IAA treatment. Note: Black lines ending with arrows and vertical lines represent promoting and inhibiting regulation, respectively. Solid and dashed lines represent the determined and undetermined interactions in the signaling pathway, respectively. Red and blue indicate differentially expressed genes that are upregulated and downregulated, respectively.

**Table 1 ijms-25-03180-t001:** *Cis*-acting elements analysis of *KN1*.

Gene Name	Name of *Cis*-Acting Element	Nucleotide Sequence	Function Annotation
*KN1*	TCA-element	TCAGAAGAGG	Salicylic acid-responsive
	CGTCA-motif	CGTCA	Methyl jasmonate-responsive
	TGACG-motif	TGACG	Methyl jasmonate-responsive
	CAAT-box	TGCCAAC/CAAAT	*Cis*-acting element shared by promoters and enhancers
	MBS	CAACTG	Drought response-related MYB binding sites
	ABRE	AACCCGG	Abscisic acid-responsive
	TATA-box	TATA	Core promoter element
	CCAAT-box	CAACGG	MYBHv1 binding site
	O2-site	GATGATGTGG	Zein metabolism-related *cis*-acting element
	A-box	CCGTCC	*Cis*-acting regulatory element
	CAT-box	GCCACT	Meristem-related *cis*-acting element
	TGA-element	AACGAC	Auxin-responsive
	GARE-motif	TCTGTTG	Gibberellins-responsive
	P-box	CCTTTTG	Gibberellins-responsive
	TCCC-motif	TCTCCCT	Light-responsive
	GATA-motif	AAGGATAAGG	Light-responsive
	Sp1	GGGCGG	Light-responsive

**Table 2 ijms-25-03180-t002:** Field phenotypes and grain traits of B73 and *kn1*.

Field Phenotype and Grain Traits	B73	*kn1*
Plant height (cm)	250.39 ± 0.87 ^a^	203.00 ± 1.15 ^b^
Ear height (cm)	108.78 ± 1.95 ^a^	93.67 ± 0.73 ^b^
Stem diameter (mm)	22.29 ± 0.38 ^a^	21.3 ± 0.66 ^a^
Chlorophyll content	50.14 ± 1.64 ^a^	52.5 ± 0.45 ^a^
Leaf length (cm)	78.54 ± 2.08 ^a^	79.06 ± 0.97 ^a^
Leaf width (cm)	9.23 ± 0.18 ^a^	8.30 ± 0.17 ^b^
Leaf area size (cm^2^)	545.56 ± 23.84 ^a^	492.74 ± 14.17 ^a^
Leaf direction value	5.35 ± 0.28 ^a^	6.10 ± 0.38 ^a^
Ear weight (g)	81.52 ± 6.98 ^a^	64.82 ± 0.95 ^b^
Ear length (cm)	10.87 ± 0.49 ^a^	10.73 ± 0.23 ^a^
Ear diameter (cm)	3.97 ± 0.32 ^a^	4.13 ± 0.09 ^a^
100-kernel weight (g)	16.61 ± 1.42 ^b^	21.65 ± 0.37 ^a^
Quantity of rows	26.67 ± 1.15 ^a^	16.67 ± 0.67 ^b^
Ear rows	16.67 ± 1.15 ^a^	15.33 ± 0.67 ^a^
Cob weight (g)	14.86 ± 1.80 ^a^	17.20 ± 0.49 ^a^
Cob diameter (cm)	2.77 ± 0.21 ^a^	3.10 ± 0.03 ^a^

Note: Different lowercase letters indicate that the corresponding trait is significantly different between B73 and *kn1*, *p* < 0.05.

## Data Availability

Data are contained within the article and Appendix A.

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
