# Peer review of "Phenotypic Investigation and RNA-seq of KN1 Involved in Leaf Angle Formation in Maize (Zea mays L.)"

_ijms, 2024, doi:10.3390/ijms25063180_

Round 1

Reviewer 1 Report

Comments and Suggestions for Authors

Summary

The manuscript provides a comprehensive exploration of the genetic factors influencing leaf area (LA) in maize, with a specific focus on the kn1 mutant. The study is well-structured, presenting detailed information on materials, methods, and results.

In conclusion, the manuscript represents a commendable effort in unraveling the genetic and molecular basis of leaf area regulation in maize, particularly through the exploration of the kn1 mutant. The experimental rigor, genetic analysis, and integration of transcriptomic data contribute to the scientific merit of the study. Addressing the following suggestions for improvement could further enhance the clarity and impact of the manuscript.

Major comments

1. Some terms, such as "occipital part," might benefit from clarification or standard botanical terminology for better comprehension.

2. The concluding remarks could be strengthened by providing additional insights into the broader implications of the findings. How might the identified pathways and genes impact maize breeding or crop improvement strategies?

3. It would be beneficial to consistently specify the units (e.g., μM) for ABA and IAA concentrations to enhance clarity.

4. Abbreviations like LAS and LDV are introduced without prior definition. It would be helpful to define them early in the text or provide a glossary.

5. In Figure 1B, the phylogenetic tree should be constructed using the maximum likelihood method, and bootstrap values must be indicated in the tree.

6. Ensure that the name of the kn1 mutant is consistently italicized throughout the manuscript. Review the entire document for accuracy.

7. Increase the size of Figure 4H to improve visibility and clarity.

8. Enlarge the graphs in Figure 5 as they are currently too small. Enhance the figure's size for better readability.

9. Remove the ® symbol from instances such as NanoPhotometer spectrophotometer and a Qubit RNA Assay Kit in a Qubit 2.0 Fluorometer.

10. For sections 4.7 and 4.9, provide detailed explanations of the transcriptome analyses. Describe the software used and specify the thresholds employed for analysis.

Minor comments

Abstract

Line 16: "maize wild type B73" could be improved to "wild-type maize B73."

Line 18: The phrase "exogenous phytohormones on the LA of kn1 was greater than that of B73" could be refined for clarity: "exogenous phytohormones had a greater impact on the leaf angle of kn1 compared to B73."

Line 22: Consider adding a comma after "LA" for better readability: "mechanism of KN1 involved in the formation of maize LA, and provides a theoretical basis..."

Line 23: Consider adding a comma after "mutant kn1" for better readability: "maize (Zea mays L.), mutant kn1; leaf angle; RNA-seq; exogenous phytohormones..."

Introduction

Line 29: Consider adding a comma after "animal feed [1]": "animal feed [1], and chemical raw materials [2-5]."

Line 34: Consider breaking the sentence for better readability: "tecture is beneficial to increase planting density and produce more grain with limited cultivated land resources. In maize, plant architecture is determined by many factors such as..."

Line 38: Consider adding a comma after "PH" for better readability: "appropriate LA and PH are helpful to alleviate lodging and increase leaf light capture area to improve yield, respectively."

Line 40: Consider rephrasing for clarity: "As endogenous growth signals, plant hormones play an important role in regulating plant growth and development [12]."

Line 51: The phrase "phytohormones to participate in the regulation of LA [19,20,24]." could be slightly clarified: "phytohormones and participates in the regulation of LA [19,20,24]."

Line 53: Consider rephrasing for better flow: "In addition, a recent study on the maize mutant Semidwarf3 (Sdw3) confirmed that ethylene is also involved in the regulation of LA [25]."

Line 55: "Three Amino Acid Length Extension (TALE) super-classes" might be clearer as "Three Amino Acid Length Extension (TALE) superclasses."

Line 62: Consider rephrasing for better clarity: "In Fragaria × ananassa Duch., either too high or too low expression of FvKNOX1 led to abnormal leaf phenotypes."

Line 66: "necessary to maintain normal leaf growth and development [30]." could be clarified: "necessary for maintaining normal leaf growth and development [30]."

Line 73: Consider rephrasing for clarity: "The concept of the transcriptome was first proposed in the study of yeast cell characteristics during different cell cycles [35]."

Line 89: "present, the regulation of KNOX TFs on LA has been studied in maize and rice" might be clarified as "present, the regulation of KNOX TFs on LA has been studied to some extent in maize and rice [38], but it is relatively rare compared with bHLH and other TFs..."

Results

Line 29: Consider adding a comma after "staple food": "staple food, animal feed [1] and chemical raw materials [2-5]."

Line 104: "changed from C to T" could be clarified for better readability: "changed from 'C' to 'T'."

Line 106: "shorten the protein product" could be clarified: "shorten the protein product, ultimately resulting in phenotypic differences between the mutant and the wild type."

Line 115: Consider rephrasing for clarity: "the occipital part of the second leaf of WT B73 at V3 stage was detected by qRT-PCR."

Line 141: "LA and PH of kn1 was significantly smaller than that of WT" could be clarified: "LA and PH of kn1 were significantly smaller than those of WT."

Line 147: Consider rephrasing for clarity: "The phenotypic differences caused by mutations were manifold."

Line 164: Consider rephrasing for clarity: "exogenous ABA concentration gradually rising to 10 μmol/L, the LA of B73 and kn1 showed a trend of first decreasing and then increasing and gradually increasing, respectively."

Line 175: "indicating that kn1 was more sensitive to exogenous plant hormones than B73" could be clarified: "indicating that kn1 exhibited greater sensitivity to exogenous plant hormones than B73."

Line 195: Consider rephrasing for clarity: "Differentially expressed genes in all samples were screened by setting the threshold of '|log2Foldchange| > 1, q < 0.05'."

Line 224: "137 (101 up-regulated and 36 down-regulated)" could be clarified: "137 DEGs (101 up-regulated and 36 down-regulated)."

Line 238: "flavone and flavonol biosynthesis" could be clarified: "flavonoid and flavonol biosynthesis."

Line 288: "The verification results showed that the expression pattern" could be clarified: "The verification results showed that the expression patterns."

Line 297: Consider rephrasing for clarity: "In order to determine the correlation between changes in transcriptome expression levels obtained by RNA-seq and LA phenotype data, DEGs conforming to the condition of 'FPKM ≥ 10' were screened from the RNA-seq data for WGCNA analysis to mine hub genes."

Line 311: Consider rephrasing for clarity: "there were seven genes with important regulatory roles, namely hub genes, in the interaction network constructed based on genes in the blue module."

Line 315: "nucleus, cell membrane components" could be clarified: "nucleus, cellular membrane components."

Discussion

Line 331: Consider rephrasing for clarity: "LA is an important factor affecting maize yield, and smaller LA is a crucial feature required for ideal maize plant architecture."

Line 332: Consider rephrasing for clarity: "At present, genes associated with maize LA are mainly involved in leaf morphogenesis and phytohormone signal transduction."

Line 335: "serine/threonine protein kinase [39]" could be clarified: "serine/threonine protein kinase (LGN) [39]."

Line 336: Consider rephrasing for clarity: "LG3 and LG4 both encode KNOX TFs [40], and LG1 and LG2 encode SPL [41] and basic leucine zipper (bZIP) [42] TFs, respectively."

Line 342: "contraction of LA of mutant kn1" could be clarified: "reduction in LA of mutant kn1."

Line 348: Consider rephrasing for clarity: "IAA and ABA are important regulatory phytohormones involved in a series of life activities, from growth and development to stress response [43,44]."

Line 352: "aminates IAA" could be clarified: "converts IAA."

Line 356: "leaf inclination by inhibiting the expression level of its target gene OsTIR1 in rice, thus determining the formation of LA in rice [15]" could be rephrased for clarity.

Line 370: "participate in the signal transduction process of various phytohormones, such as IAA (ZmARF22 and ZmIAA26), BRs (ZmBZR3), ABA (two ZmWRKY-like genes), jasmonic acid (JA) (ZmCYP75B1 and ZmCYP93D1) and ethylene (ZmABI3)" could be clarified for better readability.

Line 381: "core factors affecting the formation of LA" could be clarified: "core factors influencing LA formation."

Line 397: "IAA and BR signals and the activity of polygalacturonase were all enhanced in B73 after treatment with exogenous ABA (Figure 10A)" could be rephrased for clarity.

Line 405: "flavonoid biosynthesis of B73 and kn1 was hindered" could be clarified: "flavonoid biosynthesis was hindered in B73 and kn1."

Materials and methods

Line 431: "After that, M1 and B73 were backcross for 2 generations and self-cross for 1" - Consider rephrasing for clarity: "Afterward, M1 and B73 underwent two generations of backcrossing and one generation of self-crossing."

Line 434: "row width, row width and plant spacing" - Repetition of "row width." Consider rephrasing: "row width, inter-row spacing, and plant spacing."

Line 437: "After pollination, 3 plants with good growth and similar growth were selected" - Repetition of "growth." Consider rephrasing: "After pollination, 3 plants with robust and similar growth were selected."

Line 445: "calculate leaf area size (LAS) and leaf direction value (LDV)" - It might be helpful to define the abbreviations (LAS and LDV) before introducing them.

Line 455: "Phylogenetic tree was generated via the Neighbor-joining method with select Pairwise deletion, the value of the bootstrap was set to 1000." - Rephrase for clarity: "A phylogenetic tree was generated using the Neighbor-joining method with pairwise deletion; the bootstrap value was set to 1000."

Line 460: "Young leaves of WT B73 and mutant kn1 at V3 stage were collected from the field" - Specify the V3 stage; for example, "Young leaves of WT B73 and mutant kn1 at the V3 growth stage were collected from the field."

Line 471: "The PCR reaction system (20 μL) including DNA template (1 μL), the forward and reverse primers (1 μL each), ddH2O (7 μL) and 2 Ñ… SanTaq PCR Mix (10 μL)." - Consider rephrasing for clarity: "The PCR reaction system (20 μL) included 1 μL DNA template, 1 μL each of the forward and reverse primers, 7 μL ddH2O, and 10 μL 2x SanTaq PCR Mix."

Line 506: "The occipital part of the second leaf" - Clarify the term "occipital part"; it may be more commonly referred to as the "tip" or "apical portion."

Line 524: "Genes meeting the requirements of |log2Foldchange| > 1, q < 0.05 and were defined as differentially expressed genes (DEGs)." - Rephrase for clarity: "Genes meeting the criteria of |log2Foldchange| > 1 and q < 0.05 were defined as differentially expressed genes (DEGs)."

Line 534: "WGCNA package in R [63] was used to perform co-expression analysis based on RNA-seq data." - Specify the version of the WGCNA package or note that the version may change over time.

Conclusions

Line 546: "resulting in a shortened protein, and eventually caused a series of phenotype differences including PH and LA." - Rephrase for clarity: "resulting in a truncated protein, which eventually led to a series of phenotypic differences, including plant height (PH) and leaf area (LA)."

Line 548: "10 μmol/L ABA and 100 μmol/L IAA" - Specify the units for ABA and IAA concentrations for consistency (e.g., μM).

Line 552: "suggesting that KN1 may be involved in the above pathways." - Specify the pathways mentioned.

Line 554: "significantly related to cell membrane components, cellulose biosynthesis and cell wall formation." - Specify the key genes identified.

Line 556: "This study suggests that KN1 may be related to cell wall-mediated changes in LA and provides new insights into the formation of LA in maize." - Clarify the specific insights provided by the study.

Reviewer 2 Report

Comments and Suggestions for Authors

In this manuscript Wu and colleagues analyze the effects of IAA and ABA treatment on a KNOTTED1 maize mutant with respect to Leaf Angle on morphologic and RNAseq approaches identifying a different response with respect to wild type B73. This work is interesting as it shed light on the role of phytohormons on kn1 gene.

First of all it is not clear to me why you used ABA instead of other hormones with a known role in leaf angle formation: perhaps because of its synergistic role with BR? In addition, you did not introduce it anywhere in the introduction: you should. At least in order to explain why you decide to use it. In the discussion you introduce ABA somehow but, again, you have to better clarify why you decided to use it rather than, for instance, BR.

Other than this crucial point, below you can find my comments:

- section 2.1: please report the CDS position of the missense mutation you identified in the kn1 mutant to be able to localize it in the protein and the kn1 gene name in the RefGen_v4 annotation.

- section 2.2: as your mutant has been produced via EMS, it is reasonable to think that the mutation load would be high and that that in kn1 gene could not be the only one. If it is true, from one side, that you could assess other differences by using the RNAseq data you produced in this work, if you do not have other supporting data, it could be sufficient just to compare your data with those present in the literature to assess the (phenotipic) behaviour of your mutant with other similar ones. In fact, there could be the possibility that other mutations could concur to the data you observed. Several generations of backcrossing are usually required to obtain a reasonably "clean" mutant (Hasan, 2015).

- section 2.4: please confirm in the text the amount of hormones the plants were treated with.

- in the discussion you build a model based on differential expression of some genes, but I could not find anything in the results section: you could add perhaps a supplementary table indicating DEGs pinpointing at those that you deem to be more important. More in general you have to present in the results section all the data you use further in the discussion (and in your case in the model).

- knox1 gene has been proven to have multiple roles, among which in controlling GA catabolism (Bolduc, 2009) and auxin (Bolduc, 2012): I would stress this direct role a bit more clearly in your introduction and discussion

- section 4.1: please describe wt and then its mutant

- it is not clear (lines 430-431) how many  backcrosses and selfcrosses you performed to obtain the final mutant: 2+1 or more? This part has to be described properly in order to assess how "clean" your mutant is (see above)

- you reported that M1 mutants were used to mine variant sites (line 432) but you did not report any analysis to this extent: please remove it.

- section 2.6: you both watered and sprayed your plants with both hormones? Please add references

- Figure 1A: please report CDS position rather than gene position; 1C: please explain what the letters on the histogram mean.

- Figure 4: panels A to G are not necessary, please move them to Supplementary if needed and leave just panel H.

- Figure 8: this figure refers to which condition of B73 vs kn1 (with  or without phytohormones treatment) or what else? panel C and D: please enlarge text 

- Table 2: please use superscript to add letters related to statistics

Comments on the Quality of English Language

Just read thoroughly your manuscript again to look for typos and other few errors (ex.: "were backcross-ed", line 430)

Round 2

Reviewer 1 Report

Comments and Suggestions for Authors

Authors have diligently incorporated the revisions based on feedback from the reviewer. I highly recommend this manuscript for publication in its current form. Congratulations on the excellent work!

Reviewer 2 Report

Comments and Suggestions for Authors

I thank the authors for having answered most of my comments; however I fell there are some that are still pending.

The introduction has to introduce the main theme of the manuscipt, that in this case is to evaluate the roles of ABA and auxins. For this reason I asked the authors to provide some background information about both hormones. If it is true that ABA has no known role about Leaf Angle, you should at least introduce why you decided to assess it (as this idea did not came without reasons). In your response to my comment, you described the rationale of that: you should do it in the introduction as well. I suggest to modify the sentence "In conclusion, there is no evidence that ABA is directly or indirectly involved in the regulation of leaf angle, so we chose ABA in the experiment of exogenous phytohormone spraying" because it would make no sense: It does sense, instead, that you tested ABA since there are indictions that it may be involved due to the presence of ABA-responsive elements in the promoter and to its role with BR. Otherwise it is not clear why you decide to use ABA. I let you note that the term "ABA" is used only twice in the introduction, in the same sentence, and there is nothig about the reason why you decided to use it. On the other side, many more info are about the role of GA, although you did not use it.

Similarly, I expect the role and the effect of ABA in LA to be better discussed

Round 3

Reviewer 2 Report

Comments and Suggestions for Authors

I thank the authors for this last addition. My concern was that the use of ABA was not sufficiently "explained" in the introduction so that some readers may have wondered about its use.

I do not have anything else to write and I wish the best to the authors